# Multi-Level Stochastic Latent Noise Perturbation for Single Domain Generalization

## Abstract

Single domain generalization (SDG) is challenging because models trained on a single domain often suffer from out-of-distribution (OOD) shifts at inference time. Existing augmentation techniques often sacrifice semantic consistency for diversity or vice versa, and are largely confined to vision tasks. We propose a Stochastic Latent Noise Perturbation Module (SLNP) that automatically computes multiple maximum mean discrepancy thresholds based on the source domain's intra- and inter-class statistics, and then maximizes the sum of noise under these adaptive bounds. This unified objective generates diverse yet semantically faithful samples, applied independently of the downstream training loop without requiring adversarial training or auxiliary loss terms. In addition, SLNP complements normalization methods, yielding synergistic improvements when the two are combined. Furthermore, our method is modality-agnostic and applicable to any distribution-based data. Experiments on image benchmark demonstrate that our approach integrates easily into existing pipelines and improves state-of-the-art SDG baselines, and additional results on speech data show its applicability beyond the vision domain.

## 1 Introduction

One of the key goals in machine learning is the learning algorithms' ability to generalize to unseen samples. The target of generalization is usually test instances, but in this work, we aim for *domain* generalization. Domain generalization is a task that seeks to transfer knowledge gained in the source domain to other related, but different, target domains (Muandet et al., 2013). Here, the concept of 'domain' describes the nature of data representation, such as image styles (photo vs. sketch vs. comic) or voice background (noiseless vs. noisy) (Li et al., 2017; Narayanan et al., 2018). In particular, we tackle the single-domain generalization problem in this work, where the prediction model is trained on a single source domain. The main objective in single-domain generalization is to enlarge diversity to cover unseen target shifts, while preserving semantic consistency. Recent works have begun to balance this diversity vs. semantic consistency trade-off, but the majority of them are restricted to the vision tasks, leaving the multi-modal approach largely unexplored (Wang et al., 2021; Choi et al., 2023; Zheng et al., 2024).

In real-world applications, the single-domain constraint naturally arises due to data scarcity, privacy concerns, and high costs, and this challenge is not confined to vision tasks but is equally relevant in speech and other modalities. Models trained in such settings are often required to face unseen domains during test time. For example, in autonomous driving, the training data may only cover a limited range of weather conditions or a single geographic region (Sanchez et al., 2023). But when the system is deployed, it suddenly needs to deal with rain, snow, or roads that look different from the training set (Qi et al., 2024). The same kind of issue shows up in speech recognition. A model might be built using recordings from one device or one quiet environment, and later it is expected to work under very different acoustic settings (Kim et al., 2022b). These situations suggest that single-domain generalization cannot be seen only as a benchmark exercise. In practice, it shows up as a recurring difficulty.

To deal with single domain generalization, existing approaches demonstrated effectiveness on vision only or speech only benchmarks. But since they are inherently tied to modality-specific structure and statistics, it is difficult to transfer them to other modalities. To address these limitations, we propose a new augmentation framework that is classifier-independent, semantic-preserving, and

modality-agnostic. Yüksel et al. (2021) explored latent-space perturbations with normalizing flows, showing that invertible mappings can provide on-manifold variations. They consider randomized and adversarial variants, but the closeness is enforced only in latent space. In contrast, our method perturbs latent representation through a flow-based model under a multi-level Maximum Mean Discrepancy(MMD) constraint derived from domain-specific statistics in the image space, controlling semantic fidelity at the perceptual level. This maximizes diversity while preserving class semantics, and functions as a modular component transferable across modalities, from images to waveforms, providing a unified solution for single domain generalization.

The main contributions of this work can be summarized as follows:

- We introduce a modular augmentation method that operates without end-to-end adversarial training, expanding diversity while preserving semantics through an MMD-based constraint.
- Our framework directly transfers to different data modalities (e.g., images and speech), enabling a modality-agnostic perspective on single-domain generalization.
- We demonstrate strong performance on both vision (PACS, CIFAR-10-C) and speech (TAU Urban Acoustic Scenes) datasets, showing that our approach complements normalization-based methods and achieves competitive or superior accuracy compared to recent SDG baselines.

## 2 RELATED WORK

**Multi Source Domain Generalization** Domain generalization aims to build models that perform well on unseen target domains. Early studies such as Volpi et al. (2018) approached this challenge by learning domain-invariant representations through adversarial data augmentation to generate worst-case perturbations around source distributions, and Zhao et al. (2020) later introduced meta-learning frameworks that episodically split source domains into meta-train and meta-test subsets, combined with entropy regularization, to better simulate domain shifts. Normalization-based methods by Seo et al. (2020) adapt feature statistics, optimizing domain-specific normalization layers. Zhou et al. (2021) and Li et al. (2021) suggest data augmentation as an effective approach, including instance-level style mixing and simple feature perturbations. These methods have demonstrated strong results in multi-domain settings; however, most approaches generally rely on the existence of multiple source domains and therefore cannot be directly applied or exhibit poor performance when applied to single-source domain generalization.

**Single Domain Generalization** Single-domain generalization was introduced by Qiao et al. (2020), a meta-learning framework that generates pseudo-domains via style perturbations within the source data to simulate domain shifts without requiring multiple sources. In vision tasks, Wang et al. (2021) learns augmentation patterns directly from the source domain using a stylization module. More recent vision-specific single-domain generalization methods focus on balancing diversity with semantic preservation. Zheng et al. (2024) leverages learnable semantic transformations with standard image augmentation operations such as contrast and rotation. In another line of work, Zhou et al. (2021) generated diverse features by mixing instance-level styles, while Xu et al. (2021) and Choi et al. (2023) applied random convolutional filters to diversify feature statistics. Furthermore, Liu et al. (2024) combined stylization and destylization modules within an adversarial framework to improve semantic preservation in an end-to-end manner, and Efthymiadis et al. (2025) introduced an artificial validation set generated from transformed source data to guide augmentation design.

While research on single-domain generalization (SDG) in vision tasks has been more active, work on speech data has been relatively limited. Nevertheless, single-domain generalization in speech datasets has been investigated in several tasks where domain shifts arise from recording conditions or signal processing variability. In the acoustic scene classification task, the DCASE 2021 Challenge Task 1A highlighted the difficulty of generalizing across devices, and the winning system employed Residual Normalization to reduce device-specific biases (Kim et al., 2022a). This setting of DCASE 2021 aligns well with the domain generalization problem, as it can be reformulated into a single-source setting by training on one device and evaluating on unseen devices. By contrast, later DCASE challenges shifted their focus toward efficiency, imbalance, and data-limited learning, which are important but do not directly correspond to our single domain generalization scenario.

A subsequent method, Relaxed Instance Frequency-wise Normalization (RFN), extended Residual Normalization with instance-level frequency normalization and a relaxation mechanism, achieving improved robustness on the TAU Urban Acoustic Scenes 2020 Mobile dataset (Kim et al., 2022b). Beyond scene classification, single-domain generalization has also been explored in audio deepfake detection, where methods are designed to generalize across unseen spoofing algorithms and vocoder artifacts through an audio-specific module (Xie et al., 2023).

Although extensive research has been conducted on single-domain generalization within individual modalities such as vision and speech, these methods are tied to modality-specific assumptions, which limit their applicability across different data modalities.

**Modality Agnostic Single Domain Generalization** Uncertainty-Guided Generalization method was the first to explicitly developed for modality agnostic single-domain generalization, leveraging uncertainty estimation within a Bayesian meta-learning framework to guide augmentation in both the input and label spaces (Qiao & Peng, 2021). Modality-Agnostic Debiasing (MAD) separates domain-specific from domain-invariant information through a dual-branch architecture, achieving generalization gains across modalities (Qu et al., 2023). However, follow-up research has largely diverged into vision-only or speech-only directions, leaving the multi-modal objective unfulfilled. Moreover, augmentation-based modality-agnostic approaches for SDG remain unexplored, as domain shifts differ significantly across modalities. Our method addresses this gap by applying distribution-based augmentations in a modality-agnostic manner across vision and speech datasets. By combining a Stochastic Latent Noise Perturbation (SLNP) module with existing modality-specific normalization strategies, we achieve semantically consistent yet diverse augmentations that adapt naturally to distributional biases across different modalities.

**Complementarity of Augmentation and Normalization** In single-domain generalization, augmentation and normalization have emerged as two major strategies. Augmentation mitigates the limitation of training on a single source by generating pseudo-domains that enhance diversity and improve robustness against unseen domains (Volpi et al., 2018; Zhou et al., 2021; Zheng et al., 2024). However, augmentation alone often sacrifices semantic consistency, as perturbed samples may deviate in ways that enlarge distribution gaps. Normalization suppresses or aligns these domain-specific biases in the feature space, yielding more reliable domain-invariant representations (Ioffe & Szegedy, 2015; Seo et al., 2020; Lee et al., 2023). Yet normalization doesn't provide the diversity needed to cover the target shifts. These complementary strengths suggest that combining augmentation and normalization is a promising direction. Augmentation introduces diversity, while normalization projects these diverse features into a shared space that stabilizes semantics.

Building on this intuition, Fan et al. (2021) complements adversarial domain augmentation with a learned normalization module that adapts standardization and rescaling to incoming domains. While adversarial domain augmentation adversarially perturbs the source distribution to synthesize challenging pseudo-domains, it doesn't include an explicit semantic-preserving constraint, which may alter class relevant feature. Liu et al. (2024) adopts a different strategy by combining stylization and an adversarially trained destylization module in a min–max framework, further reinforced with a semantic consistency loss. This design explicitly encourages semantic preservation at the representation level. While both methods demonstrate the benefit of coupling augmentation with normalization, their portability remains limited because each relies on augmentation mechanisms that are tied to the training pipeline. ASR-Norm (Fan et al., 2021) depends on adversarially generated pseudo-domains produced by ADA (Volpi et al., 2018), which is designed to create worst-case distributions rather than preserve semantics. Likewise, StyDeSty (Liu et al., 2024) requires an end-to-end min–max framework with a joint stylization–destylization objective, making its augmentation tightly coupled with the classifier. Consequently, neither approach provides a plug-and-play semantic-preserving transformation that can be easily reused in other pipelines.

In contrast, our method is designed to be semantic-preserving from the outset. We perturb primarily domain-specific factors under a Maximum Mean Discrepancy (MMD) constraint, encouraging semantic structure to remain intact while expanding diversity. Unlike ADA or stylization-based approaches, our method does not rely on adversarial recovery or specialized modules, making it a lightweight and modular solution. This allows normalization to focus solely on eliminating residual domain biases without risking semantic degradation. The resulting synergy enhances semantic stability against style fluctuations, yielding domain-invariant representations that can serve as a plug-and-play augmentation beyond end-to-end frameworks.

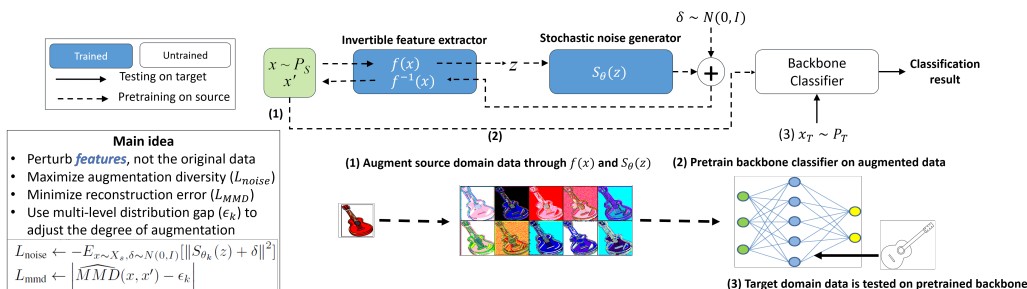

Figure 1: Stochastic Latent Perturbation Module (SLNP) Framework

## 3 PROPOSED METHOD

Single domain generalization aims to learn a robust model from a single source domain $\chi_s = \{(x_i, y_i)\}_{i=1}^N$, $x_i \sim P_s$, $y_i \in \{1, .., C\}$ where $x_i \sim P_s$ are drawn from the source distribution $P_s$, $y_i \in \{1, ..., C\}$ are class labels from $C$ categories, and $N$ is the number of training examples. The goal is to generalize to previously unseen target domain examples $x \sim P_t$, where $P_t$ denotes the target domain distribution and satisfies $P_s \neq P_t$. While the source examples and target examples are drawn from different distributions, the label space $y \in \{1, ..., C\}$ remains the same. To bridge the distribution shift between $P_s$ and $P_t$, augmented data $x^+$ can simulate potential variations in $P_t$ while preserving the semantics of $P_s$.

We introduce a Stochastic Latent Noise Perturbation (SLNP) module, a sampling method that enlarges the training set by applying a non-linear latent space transformation to the source data $x$ under multiple MMD constraint thresholds. Specifically, we encode the input using an invertible flow to obtain the latent features, perturb them with a stochastic noise tensor and decode the result back into the valid image space with inverse flow. For each MMD threshold $\epsilon_k$, where $k \in \{1, ..., K\}$ indexes the set of $K$ thresholds, we optimize a loss that maximizes the noise magnitude while enforcing the MMD between original and perturbed samples to match $\epsilon_k$. These thresholds are automatically derived from the source domain. Once trained, the module operates independently of the network and can be used as a plug-in data augmentation method for both image and speech data.

### 3.1 STOCHASTIC LATENT PERTURBATION MODULE

We adopt a RealNVP-style encoder, decoder consists of affine coupling layers and a learnable global scale (Dinh et al., 2017). RealNVP introduces invertible transformations using coupling layers that split the input into two parts, where one part remains unchanged while the other is updated through a scale-and-shift transformation predicted from the unchanged part. This design enables exact inversion and efficient Jacobian computation, making the model suitable for stable feature manipulation. The flow isn't trained to maximize the likelihood, but the structure simply serves as an invertible encoder, so that the perturbed data can remain in a semantically consistent manifold. Our variant keeps only the components required for inversion and integrates them with the noise perturbation objective.

For RGB images, we use a RealNVP-style flow with an asymmetric channel split per coupling layer. The last channel is used as the conditioner and the first two channels are transformed. Each layer predicts a 2 channel shift and a 1 channel log-scale from the conditioner, and the log-scale is clamped and broadcast to the 2 transformed channels. Since the outputs are concatenated as $[y_1, y_2]$ and the next layer again splits the channels in the same manner, the conditioning role rotates across layers and no channel remains permanently fixed. A learnable per-channel global scale is applied at the end of the flow, and semantic noise is injected in the latent space. The per-channel global scale compensates for the clamped coupling log-scales by restoring proper latent magnitude, ensuring balanced noise injection across channels.

For speech, we operate directly on the waveform with a 1D RealNVP-style coupling flow that splits along the temporal axis into even and odd samples. The even part conditions the affine transform applied to the odd part with a clamped scale. We suggest temporal splitting since frequency rep-

resentations entangle both domain-specific factors and semantic content, and perturbing frequency risks corrupting task-relevant cues. Instead, we insist that temporal perturbations yield localized changes that preserve global spectral structure. We include invertible $1 \times 1$ convolution between coupling blocks for a scalar gain. The perturbed waveforms are converted to log-mel spectrograms and passed to the acoustic backbone.

We treat the number of coupling layers and the scaling clamp range as hyperparameters. In practice, we use 4 layers for images and speech modeling, chosen to balance transformation capacity and computational stability.

Given an input mini-batch $x$, which can be of any modality, we first obtain a latent feature map $z = f_\phi(x)$ through the flow encoder. Stochastic perturbation function then produces the noise tensor.

$$\varepsilon = S_\theta(z) + \delta, \quad \delta \sim N(0, I) \tag{1}$$

The noise tensor consists of a deterministic term and a stochastic term. $S_\theta$ is a simple 2-layer 3x3 Convolutional Neural Network and $\delta$ is independently drawn from $N(0, I)$. The random term is resampled at every forward pass, and this stochasticity enlarges the training distribution effectively and prevents the classifier from overfitting to the deterministic noise pattern.

By decoding it through an invertible mapping, our method generates diverse augmentations. $\alpha$ is a hyperparameter for scaling the noise tensor.

$$x' = f_\phi^{-1}(f_\phi(x) + \alpha \cdot \varepsilon) = f_\phi^{-1}(z + \alpha \cdot \varepsilon) \tag{2}$$

Specific details of the hyperparameters and the architectures of the flow-based models for both image and waveform datasets are provided in Appendix.

## 3.2 OBJECTIVE

We train the flow $f_\phi$ and the perturbation module $S_\theta$ with the objective below, doing so separately for each MMD threshold $\epsilon_k$. For each $\epsilon_k$, we jointly optimize the flow parameters $\phi$ and the perturbation module parameters $\theta$ by minimizing the loss, where $k \in \{1, ..., K\}$ indexes the distinct MMD thresholds.

$$L_k(\phi, \theta) = -\lambda_1 E[\|\varepsilon\|^2] + \lambda_2 |\widehat{MMD}(x, x') - \epsilon_k| \tag{3}$$

Here, $\widehat{MMD}(x, x')$ is a shorthand notation for the empirical MMD between the two mini-batches $x$ and $x'$. The first term in Eqn 3 forces the module to push augmented samples away from the source by injecting a large noise into the latent space. The second term counterbalances this expansion by penalizing the distributional gap relative to the target threshold, projecting the samples back so that the empirical MMD nearly matches $\epsilon_k$. Together, these two terms jointly balance the trade-off between maximizing the variability and preserving semantics. This yields K distinct pairs $\{f_{\phi_k}, S_{\theta_k}\}_{k=1}^K$. The number of MMD thresholds K is a tunable hyper-parameter that controls the range of allowable distribution shifts. The sequence $\{\epsilon_k\}$ is designed to decrease over $k$ in our experiments, allowing for larger MMD gaps in the beginning and closing the difference over time. The rationale is to progressively increase the difficulty level of optimization, similarly to score-based diffusion models (Song et al., 2021) and curriculum learning (Soviany et al., 2022).

$\epsilon$ - **list Construction:** Leaving the range of $\epsilon_k$ $(k = 1, ..., K)$ as a hyperparameter is risky, because it can produce distribution shifts that are either too weak or too aggressive, and it is difficult to manipulate. To eliminate this uncertainty, we determine the range automatically from two distribution-specific statistics. We computed (1) Minimum inter-class MMD distance $\xi_{inter}$ and the (2) average intra-class dispersion $\xi_{intra}$, both measured with the same RBF-kernel function $k(\cdot, \cdot)$. Here $k$ is the kernel function, not the MMD-level index used in Eq. (3). Here, $x$ and $x'$ denote individual samples drawn from the same class when computing intra-class dispersion $E_x[k(x, x)] - E_{x, x'}[k(x, x')]$, where the expectation is with respect to the samples belonging to the same class[1]. This definition is distinct from Eq. (3), where $x$ and $x'$ denote mini-batches. We then define the maximum admissible MMD threshold $\epsilon_{max} = \frac{\xi_{inter}}{2\xi_{intra}}$. This ensures that augmented examples remain, on average, closer

---

[1] Details in appendix.

---

**Algorithm 1** Stochastic Latent Noise Perturbation Module (SLNP) Pretraining

---

**Require:** Source dataset $\chi_s$, number of MMD levels $K$, MMD thresholds $\{\epsilon_1, ..., \epsilon_K\}$, number of training epochs $T$

1: **for** $k = 1$ **to** $K$ **do**
2:      Randomly initialize flow $f_{\phi_k}$ and perturbation generator $S_{\theta_k}$
3:      **for** $t = 1$ **to** $T$ **do**
4:          Sample mini-batch $x \sim \chi_s$
5:          $z \leftarrow f_{\phi_k}(x)$
6:          $\varepsilon \leftarrow S_{\theta_k}(z) + \delta, \qquad \delta \sim N(0, I)$
7:          $x' \leftarrow f_{\phi_k}^{-1}(z + \alpha \cdot \varepsilon)$                              ▷ Compute loss
8:          $L_{\text{noise}} \leftarrow -E_{x \sim X_s, \delta \sim N(0,I)}[\|S_{\theta_k}(z) + \delta\|^2]$
9:          $L_{\text{mmd}} \leftarrow \left| \widehat{MMD}(x, x') - \epsilon_k \right|$
10:         $L_k(\phi_k, \theta) \leftarrow \lambda_1 \cdot L_{\text{noise}} + \lambda_2 \cdot L_{\text{mmd}}$
11:         Update $f_{\phi_k}$ and $S_{\theta_k}$ to minimize $L$
12:      **end for**
13: **end for**

---

to their own class than to the nearest other class. Setting $\epsilon_{max}$ as the upper bound, we construct $\epsilon$-list as a sequence of K progressively smaller thresholds by uniform linear spacing.

$$\epsilon_k = \epsilon_{max} \frac{K - k + 1}{K}, \qquad k = 1, ..., K \tag{4}$$

Since both $\xi_{inter}$ and $\xi_{intra}$ are estimated directly from the input data, $\{\epsilon_k\} = \{\epsilon_1, ..., \epsilon_K\}$ is determined automatically.

### 3.3 TRAINING PIPELINE

SLNP is first pre-trained on the entire source domain, independent from the subsequent training and testing loops. The overall pre-training pipeline is summarized in Algorithm 1. Once this pre-training is done, every mini-batch is passed through the frozen perturbation module to generate additional noise-enhanced views for the backbone classifier during training. The augmented data was blended with the raw data to avoid excessive deviation from the original. By shifting the perturbation learning process outside the main training loop, we keep the classifier training lightweight while still supplying diverse, semantically faithful variants. It can also be plugged in as a data augmentation method to many other methodologies.

## 4 EXPERIMENT

Our augmentation module is trained independently from the downstream model and can be easily integrated into various learning pipelines. Our design has advantages in terms of reusability and broad applicability to any distribution-based modalities. To demonstrate the generality and effectiveness of our method, we conduct experiments by (1) integrating our augmentation module into current SDG state-of-the-art method, StyDeSty, and (2) applying our method to Speech single domain generalization task.

### 4.1 DATASETS

PACS and CIFAR-10-C are widely used vision datasets to demonstrate the effectiveness of classification models in SDG. We utilized the TAU Urban Acoustic Scenes 2020 Mobile dataset to evaluate the performance in speech SDG. We demonstrate compatibility in both vision and speech data, ensuring that our method applies to any distribution-based modality.

**PACS.** PACS (Yu et al., 2022) consists of 9,991 images. There are 4 domains (photo, cartoon, art painting, sketch) with 7 classes and a resolution of 224 x 224. One domain is chosen as the source domain, and others are used as the target domains.

**CIFAR-10-C.** CIFAR-10-C (Hendrycks & Dietterich, 2019) contains diverse corruptions to the CIFAR-10 dataset with 10 classes. CIFAR-10 (Krizhevsky, 2009) consists of 32 x 32 RGB images with 50,000 training data and 10,000 test data. The corruptions include weather, blur, digital, and noise, and the corruption level is from 1 to 5. The original CIFAR-10 dataset is used as a source domain, and the CIFAR-10-C dataset is used as target domain.

**TAU Urban Acoustic Scenes 2020 Mobile dataset.** TAU dataset (Mesaros et al., 2018) contains 10 second audio clips from 10 classes recorded in 12 European cities across multiple devices. All audio is resampled to 16kHz and transformed into 256 bin log-Mel spectrograms. Single device (Device A) is regarded as the source domain, and evaluated on this single device and other unseen domains (Device B,C and simulated channels S1-S6).

### 4.2 IMPLEMENTATION DETAILS

For image data, our method was integrated into the existing learning pipeline of StyDeSty (Liu et al., 2024) to demonstrate its compatibility. Excluding Stylization, auxiliary loss terms, and adversarial training components, we only incorporated the DeStylization module, implemented as an instance normalization layer that removes domain-specific channel statistics in the downstream network, together with our augmentation method.

For PACS dataset, we use ResNet-18 (He et al., 2016) as the backbone network, following common practice in domain generalization. The model is trained with a batch size of 32 using optimizer SGD with momentum 0.9. The initial learning rate is set to 0.001 and decayed by a factor of 10 at the 60th and 80th epochs. The hyperparameters are set to $K = 15$, $\lambda_1 = 1$, $\lambda_2 = 1$, and $\alpha = 5 \times 10^{-2}$.

For the CIFAR-10-C benchmark, we adopt WideResNet (16-4) (Zagoruyko & Komodakis, 2017) as the backbone, which is widely used for robustness evaluation. Training is performed with a batch size of 128 using optimizer SGD with Nesterov momentum of 0.9. The initial learning rate is set to 0.1 and scheduled using cosine annealing. The hyperparameters are $K = 15$, $\lambda_1 = 0.1$, $\lambda_2 = 1$ and $\alpha = 10^{-1}$.

Kim et al. (2022b) discovered that while in images, domain-specific biases are mainly reflected in channel statistics, in speech, they are captured in frequency statistics. To mitigate such biases in speech, we integrated our module with Relaxed Instance Frequency-wise Normalization (RFN), which effectively reduces device- and domain-dependent variations.

For TAU Urban Acoustic Scenes 2020 Mobile dataset, we adopt BC-ResNet-1 (Kim et al., 2021), a lightweight convolutional architecture tailored for acoustic scene classification. The model is trained with a batch size of 100 using optimizer SGD, with momentum 0.9. The initial learning rate is set to 0.001 and reduced by a factor of 100 every 30 epochs. The hyperparameters are $K = 3$, $\lambda_1 = 0.01$, $\lambda_2 = 1$, and $\alpha = 1$. In contrast to vision datasets, we adopt a smaller K for speech, since speech features are more vulnerable to semantic distortion and overly strong augmentation may interfere with task-relevant cues.

### 4.3 EXPERIMENT RESULTS

#### 4.3.1 RESULTS ON IMAGE SDG

Table 1 presents the classification results on the PACS dataset. While most existing methods were originally trained with a batch size of 64, our method was trained with a batch size of 32 due to memory limitations. To ensure a fair comparison, we reimplemented StyDeSty, the current state-of-the-art method on PACS, under the same setting using a batch size of 32. Our approach achieves competitive performance both as a standalone augmentation strategy and when combined with Destylization. Notably, ours alone already surpasses strong baselines (Zhou et al., 2021; Wang et al., 2021). When integrated with destylization, our method further improves the overall accuracy to 70.22%, closely matching the reimplemented StyDeSty under identical conditions. This demonstrates that semantic-preserving augmentation under an MMD constraint provides a strong complementary signal to normalization-based methods. Table 2 reports results on CIFAR-10-C under various corruption types. Here we used a batch size of 128, following standard practice in corruption robustness benchmarks. Our approach alone achieves an average accuracy of 78.47%. When combined with destylization, the accuracy further improves to 83.47%, on par with StyDeSty.

| Methods | Photo | Art | Cartoon | Sketch | Avg. |
|---|---|---|---|---|---|
| Vanila | 39.73 | 68.85 | 72.13 | 29.49 | 52.55 |
| JiGen (Carlucci et al., 2019) | 46.03 | 68.78 | 72.60 | 35.51 | 55.73 |
| MixStyle (Zhou et al., 2021) | 47.35 | 72.07 | 74.36 | 35.12 | 57.23 |
| ADA (Volpi et al., 2018) | 45.12 | 77.34 | 75.61 | 37.30 | 58.84 |
| ME-ADA (Zhao et al., 2020) | 45.89 | 76.09 | 74.71 | 36.01 | 58.18 |
| L2D (Wang et al., 2021) | 49.06 | 77.26 | 78.27 | 53.40 | 64.50 |
| ProRandConv (Choi et al., 2023) | 62.89 | 78.54 | 76.98 | 57.11 | 68.88 |
| LEAwareSGD (Zhang et al., 2025) | **65.05** | **79.17** | 77.16 | 57.78 | 69.46 |
| StyDeSty (Liu et al., 2024) | 62.46 | 78.81 | **79.77** | 59.60 | 69.37 ± 0.23 |
| Destylization Only | 47.86 | 69.49 | 77.43 | 38.68 | 58.39 ± 0.08 |
| Ours Only | 52.13 | 69.47 | 75.93 | 56.49 | 64.72 ± 1.21 |
| **Ours + Destylization** | 63.02 | 77.40 | 77.15 | **63.31** | **69.82 ± 0.36** |

Table 1: Comparison of SDG performance on the PACS dataset. Results are reported across 4 target domains (Photo, Art, Cartoon, Sketch). **Best** and second-best are highlighted.

| Methods | Noise | Blur | Weather | Digits | Avg. |
|---|---|---|---|---|---|
| Vanila | 55.02 | 73.28 | 84.40 | 61.09 | 72.83 |
| StyDeSty (Liu et al., 2024) | 76.45 | **83.43** | 87.39 | **86.75** | **83.33 ± 0.17** |
| Destylization Only | 61.79 | 81.49 | **88.30** | 84.18 | 80.13 ± 1.68 |
| Ours Only | 75.87 | 75.07 | 83.33 | 79.60 | 80.01 ± 1.53 |
| **Ours + Destylization** | **79.31** | 80.86 | 87.51 | 86.18 | 83.23 ± 0.24 |

Table 2: Comparison of SDG performance on the CIFAR-10-C dataset. Classification accuracy is shown under different corruption types (Noise, Blur, Weather, Digits).

These results confirm that semantic-preserving perturbations not only strengthen model robustness against distributional shifts but also integrate effectively with normalization-based methods. Importantly, our method can serve as a plug-and-play augmentation module, improving generalization even without adversarial recovery or specialized auxiliary networks.

Comparing our augmentation method combined with destylization against StyDeSty, we observe that both achieve similar performance on PACS and CIFAR-10-C. This suggests that the two frameworks effectively couple augmentation with normalization to balance diversity and invariance. However, we offer distinct advantages. In contrast to original StyDeSty, which requires an adversarial stylization–destylization pipeline trained end-to-end, ours provides a modular augmentation that is semantic-preserving by design. This makes it readily usable as a plug-and-play augmentation in diverse pipelines, without requiring adversarial training. At the same time, when integrated with destylization, our method consistently closes the gap with the normalization-based approaches, demonstrating that it complements such frameworks without structural overhead. A further point of distinction arises in the vision benchmarks. On PACS and CIFAR-10-C, our augmentation alone already improves accuracy beyond several state-of-the-art augmentation strategies, showing that explicitly semantic-preserving perturbations are effective even without normalization. This suggests that in visual domains, where semantic content and domain-specific style factors are relatively separable, augmentation itself can substantially enhance generalization.

### 4.3.2 RESULTS ON SPEECH SDG

Table 3 presents results on the TAU Urban Acoustic Scenes 2020 Mobile dataset, which evaluates domain generalization under both device variation (A–C) and simulated device shifts (S1–S6). For this dataset, we adopted a batch size of 100. Our augmentation method alone achieves an average accuracy of 31.82%, which is lower than the vanilla baseline, suggesting that in the speech domain, semantic and domain-specific factors are more tightly entangled and standalone augmentation may distort task-relevant cues. When combined with Kim et al. (2022b), however, our method reaches

45.19%, the strongest result among all compared methods. This demonstrates that the synergy between semantic-preserving augmentation and robust normalization is particularly important for speech single-domain generalization.

| Methods | A | B | C | S1 | S2 | S3 | S4 | S5 | S6 | Avg. |
|---|---|---|---|---|---|---|---|---|---|---|
| Vanila | 63.03 | 41.64 | 50.15 | 18.18 | 28.79 | 26.97 | 28.48 | 32.42 | 27.58 | 35.25 |
| RFN (Kim et al., 2022b) | **71.21** | 50.15 | **60.79** | 29.09 | 25.45 | 34.24 | 31.21 | 35.45 | 23.03 | 41.06 ± 3.02 |
| Ours Only | 58.79 | 43.77 | 47.42 | 18.79 | 20.00 | 26.97 | 25.15 | 27.88 | 17.58 | 32.80 ± 0.98 |
| **Ours + RFN** | 59.09 | **52.58** | 57.75 | **38.79** | **35.76** | **43.94** | **41.82** | **42.12** | **34.85** | **45.19** ± 1.12 |

Table 3: Comparison of SDG performance on the TAU Urban Acoustic Scenes 2020 Mobile dataset. Results are reported across devices (A–C) and simulated channels (S1–S6).

In contrast to the image SDG, the results on TAU Urban Acoustic Scenes 2020 shows a different trend. Using augmentation alone results in lower accuracy than the vanilla baseline; however, adding a normalization technique leads to a clear improvement. A plausible reason is that, unlike in vision, where semantics (such as shapes or edges) and style (such as color or illumination) tend to occupy different dimensions of the data, speech signals represent both in the frequency domain. As a result, device responses and channel effects are entangled with semantic cues in the spectrogram. As demonstrated by Kim et al. (2022b), frequency statistics encode strong domain-specific factors in acoustic scene recordings, which often interact with task-relevant information. Consequently, standalone augmentation may distort semantic cues along with style, leading to performance degradation, whereas normalization-based modules are necessary to suppress frequency-domain biases and restore domain-invariant structure. Nevertheless, augmentation remains crucial in SDG, even though it may underperform on its own in speech datasets; it provides the diversity that normalization alone cannot, and its combination with normalization yields complementary gains.

Overall, these results validate the claim of this work: explicitly semantic-preserving augmentation under distributional constraints can substantially improve generalization in combination with normalization. Our augmentation method operates as an independent semantic-preserving augmentation while integrating with normalization when available. Examples of augmented results in both image and speech datasets are attached in the Appendix. Despite simplified integration, our approach achieves comparable performance to state-of-the-art methods within the margin of error. This versatility makes our method broadly applicable across modalities and architectures, providing a practical and effective direction for single-domain generalization.

To better understand this effect, we analyze the roles of augmentation and normalization in single-domain generalization. In both the vision and speech domains, we observed that augmentation alone is insufficient to achieve strong generalization, as latent perturbations by themselves cannot fully bridge the domain gap introduced by domain-specific biases. Conversely, normalization methods effectively mitigate such biases, yet they fail to expose the model to sufficiently diverse unseen domains. These complementary limitations suggest that augmentation-based approaches and domain-specific normalization strategies should be applied jointly, leading to consistent and robust performance improvements in SDG.

### 4.4 ABLATION STUDY

**Hyperparameter Sensitivity Analysis of** $\lambda_1, \lambda_2, \alpha$ We conduct a parameter sensitivity analysis on CIFAR-10-C by varying both $\lambda_1$(noise magnitude maximization) and $\lambda_2$(mmd constraint). Specifically, we vary both parameters $\in \{0.05, 0.1, 1.0\}$ while keeping all other components fixed. The resulting accuracy yields a small variance of $82.78 \pm 0.50$, indicating that the proposed SLNP module is highly robust to the choice of loss balancing coefficients. Additionally $\alpha$, the scalar multiplied by the random noise is fixed throughout all experiments. It serves only as a stability factor to prevent excessive noise injection during the early stages of flow inversion.

**Sensitivity Analysis of K** We further investigated the effect of the number of perturbation levels $K$ on generalization performance using CIFAR-10-C. In this experiment, $K$ was varied from 5 to 20 in increments of 5, with all other hyperparameters fixed. We insist that the parameter $K$ balances

the semantic preservation level and diversity. While smaller values produce fewer pseudo-domains, limiting diversity, larger values provide more diverse perturbations but also increase the risk of semantic drift and training instability.

Figure 2 illustrates the model's sensitivity to the choice of $K$. We observe that performance improves as $K$ increases from 5 to 15, indicating that additional perturbation levels expose the model to a richer spectrum of domain shifts and thus strengthen robustness against corruption. However, when $K$ is further increased to 20, accuracy slightly decreases, suggesting that excessive perturbation levels may introduce redundancy or lead to the accumulation of perturbations that partially distort semantics. While the differences in accuracy don't deviate much, the best performance was achieved at $K = 15$. We find that increasing $K$ generally enhances robustness by enriching pseudo-domains, but performance saturates beyond a certain point, with $K = 15$ gives the most stable improvement.

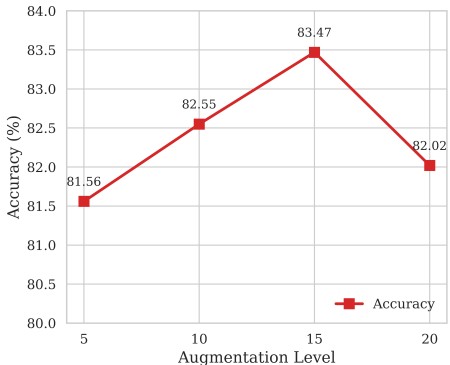

Figure 2: Sensitivity analysis on the number of augmentation samples ($K$).

## 5 CONCLUSIONS

In this work, we introduced a Stochastic Latent Noise Perturbation (SLNP) Module for single-domain generalization. By injecting stochastic noise in the latent space under multi-level MMD constraints that are automatically derived from the data, our method balances two key objectives: increasing diversity while preserving semantic consistency. The module is trained independently of the downstream classifier, making it easy to reuse, integrate into existing pipelines, and apply across modalities, from images to speech. Unlike prior approaches that are tied to modality-specific assumptions, our framework is modality-agnostic. This allows us to generate diverse samples directly from a single source domain and, when combined with modality-specific normalization strategies, achieve stronger and more reliable performance under domain shift. Across vision and speech benchmarks, we show that the proposed augmentation complements state-of-the-art SDG methods and consistently improves their generalization ability. Taken together, our work suggests a simple but effective distribution-based perturbation method that can serve as a general augmentation strategy for robust single-domain generalization.

## ETHICS STATEMENT

We use only public datasets used in several benchmarks in SDG; no private or identifiable data are used. We will release code/configurations for third-party audits to support environmentally responsible research.

## REPRODUCIBILITY STATEMENT

All results are reproducible with our code; all datasets (PACS, CIFAR-10-C, TAU Urban Acoustic Scenes 2020 Mobile) are public with download instructions. For fair baseline comparisons, we follow the official L2D and StyDeSty implementations and hyperparameters, and provide configs, seeds, and one-command scripts in the repository.

## LARGE LANGUAGE MODELS USE

Large Language Models (LLMs) were used solely to aid in writing and polishing the manuscript. Specifically, we used LLMs to improve grammar, phrasing, and clarity of exposition, without generating original ideas, experiments, or results. All technical content and experiments were designed and verified entirely by the authors.

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

## A    SAMPLE AUGMENTATIONS

We show a few sample augmentations generated by $S_\theta$ on both the image and the sound domains.

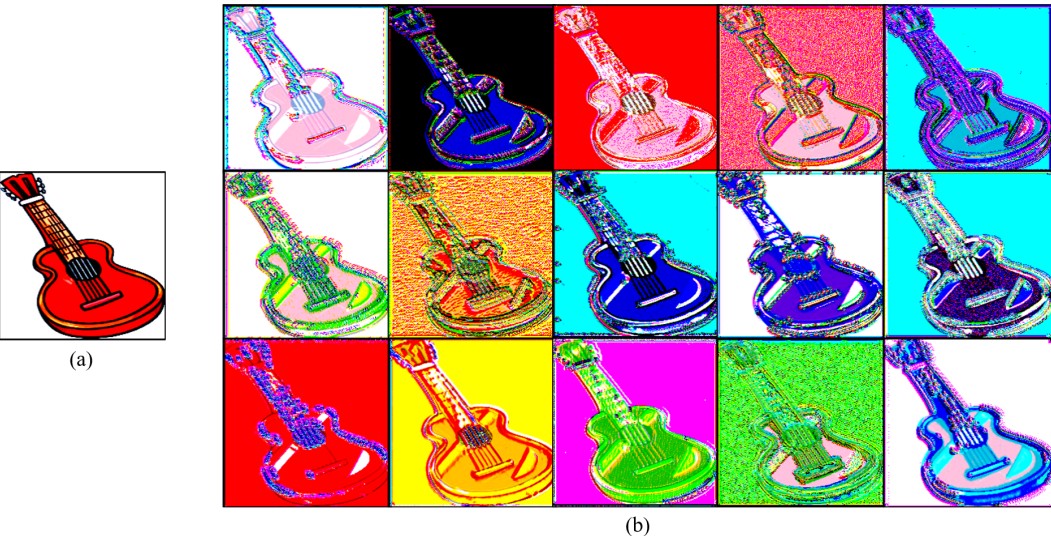

(a)

(b)

Figure 3: Examples of augmentations on the PACS dataset (Cartoon domain). (a) An original source-domain image. (b) 15 augmented samples generated by SLNP, showing diverse style variations while preserving semantic content.

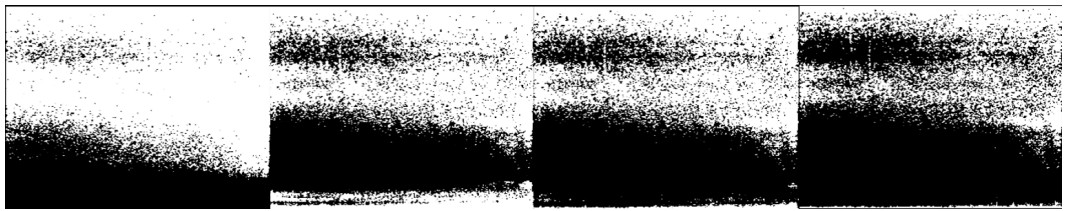

Figure 4: Examples of augmentations on the TAU Urban Acoustic Scenes 2020 dataset. The leftmost panel shows a log-mel spectrogram from the training source domain, while the remaining three panels are augmented versions generated by SLNP.

## B    THEORETICAL ANALYSIS

We justify the design of the $\{\epsilon_k\}$ by relating the augmentation bound to class separation and class spread, yielding a simple half-gap rule that keeps augmented samples within their class and is easy to enforce.

**Proposition - Definition and role of $\epsilon_{max}$** We first justify the construction of Eq 4: The minimum inter-class MMD distance is halved and normalized by the average intra-class dispersion, so that the largest allowed perturbation remains within every class boundary. Derivation of this main design is explained below.

For any class $c$, let $\xi_{inter}$ be the MMD distance to the nearest class $c'$. $P_c$ and $P_{c'}$ are the distributions those respective classes, and $P_c^{aug}$ is the distribution resulting from augmenting $P_c$.

We set the augmentation boundary as $MMD(P_c, P_c^{aug}) \leq \frac{\xi_{inter}}{2}$. Then by the triangle inequality we obtain $MMD(P_{c'}, P_c^{aug}) \geq \xi_{inter} - MMD(P_c, P_c^{aug}) \geq \frac{\xi_{inter}}{2}$. This guarantees that the

augmented distribution doesn't not cross the nearest class boundary, since it remains at least $\frac{\xi_{inter}}{2}$ away from the closest foreign class.

Building on this baseline result, we incorporate the intra-class dispersion $\xi_{intra}$ as a normalization factor. Combining the safety margin with intra-class normalization yields the final construction,

$$\epsilon_{max} = \frac{\xi_{inter}}{2\xi_{intra}}$$

To ensure that the intra-normalized perturbation still prevents boundary crossing, we must check that the triangle inequality continues to hold under this construction.

If $MMD(P_c, P_c^{aug}) \leq \epsilon_{max} = \frac{\xi_{inter}}{2\xi_{intra}}$, the triangle inequality guarantees

$$\xi_{inter} = MMD(P_c, P_{c'}) \leq MMD(P_c, P_c^{aug}) + MMD(P_{c'}, P_c^{aug})$$

$$\leq \frac{\xi_{inter}}{2\xi_{intra}} + MMD(P_{c'}, P_c^{aug})$$

$$\Rightarrow \xi_{inter}\left(1 - \frac{1}{2\xi_{intra}}\right) \leq MMD(P_{c'}, P_c^{aug})$$

Notice that even when $\xi_{inter}$ is small and $\xi_{intra}$ is large (i.e., the class boundary is uncertain), augmentation will help maintain the distance between the augmented samples and the nearest foreign samples. On the other hand, if $\xi_{intra}$ is small (i.e., samples in each class are tightly clustered), the above bound becomes very loose, and almost trivial. However, such a case also implies that the samples in that class have low level of diversity. A low value of $\xi_{intra}$ will prompt $\epsilon_{max}$ to be large, leading to more aggressive augmentation. Thus we can see that $\xi_{intra}$ automatically controls the degree of augmentation by only looking at the current dataset.

**Computing $\xi_{intra}$:** Averaging the MMD between two random halves of a class provides a way to measure the class's intrinsic dispersion in the chosen Reproducing Kernel Hilbert Space (RKHS). This measurement scales the $\epsilon$ - list so that the augmentation strength is automatically matched to the data's variability. We assume that the classes with small variability allow larger perturbations, whereas samples already scattered near a decision boundary should receive tighter noise bounds to preserve semantic consistency.

We estimate each class's intrinsic dispersion by repeatedly splitting the data into two independent subsets $X_{1c} = \{x_i\}_{i=1}^m$ and $X_{2c} = \{x_j\}_{j=1}^m$, drawn i.i.d from $P_c$. For each split, we compute the squared MMD between them, and average the result over multiple splits; the derivation is given below.

$$E\left[\widehat{MMD}^2(X_{1c}, X_{2c})\right]$$

$$= E[\|\mu_{X_{1c}} - \mu_{X_{2c}}\|_{\mathcal{H}}^2]$$

$$= E[\|\mu_{X_{1c}}\|^2 + \|\mu_{X_{2c}}\|^2 - 2\langle\mu_{X_{1c}}, \mu_{X_{2c}}\rangle]$$

$$= 2E[\|\mu_{X_{1c}}\|^2] - 2E[\langle\mu_{X_{1c}}, \mu_{X_{2c}}\rangle]$$

$$= 2E[\|\mu_{X_{1c}}\|^2 - \langle\mu_{X_{1c}}, \mu_{X_{2c}}\rangle]$$

$$= 2\left(\frac{1}{m}E_x[k(x,x)] + \left(1 - \frac{1}{m}\right)E_{x,x'}[k(x,x')]\right) - 2E_{x,x'}[k(x,x')]$$

$$= \frac{2}{m}\left(E_x[k(x,x)] - E_{x,x'}[k(x,x')]\right)$$

$$= \frac{2}{m}\text{Var}_{\mathcal{H}}(P_c)$$

$X_{1c}$ and $X_{2c}$ are independent random halves of samples from class $c$, $\mu_{X_{1c}}, \mu_{X_{2c}} \in \mathcal{H}$ are empirical mean embeddings in the RKHS, and $Var_H(P_c)$ denotes the variance of class $c$ in the RKHS. Because the two subsets are drawn i.i.d. from $P_c$, the expectations over them are symmetric. The resulting expected squared MMD between the two halves is proportional to the RKHS variance of the class. This process yields an unbiased estimator of the class's intrinsic dispersion in the RKHS.

# C COMPOSITIONS OF SLNP MODULE

## C.1 FLOW-BASED MODEL FOR IMAGE AUGMENTATION

We implement a normalizing flow based on RealNVP, tailored for RGB images ($C = 3$), to perform semantic-preserving stochastic latent perturbation. The overall augmentation module consists of an encoder-decoder flow $f_\phi, f_\phi^{-1}$, and a learnable noise generator $S_\theta$.

**2D Coupling Layer**
Given input $x \in \mathbb{R}^{B \times 3 \times H \times W}$, we split it along the channel dimension.

$$x_2, x_1 = chunk(x, 2, dim = 1),$$

where $x_1 \in \mathbb{R}^{B \times 1 \times H \times W}, x_2 \in \mathbb{R}^{B \times 2 \times H \times W}$ $x_2$ be a transformed part conditioned by the conditioning part $x_1$.

The affine transformation parameters be computed as

$$[shift, log\ scale] = chunk(f(x_1), 2,\ dim = 1)$$

$$shift \in \mathbb{R}^{B \times 2 \times H \times W},\ log\ scale \in \mathbb{R}^{B \times 1 \times H \times W}$$

Then,

$$scale \leftarrow exp(clamp(log\ scale, -7, 7))$$

The transformed output becomes

$$y_1 = x_1$$
$$y_2 = x_2 \odot scale + shift$$
$$y = [y_1, y_2]$$

This ensures invertibility

$$x_1 = y_1$$
$$x_2 = (y_2 - shift) \oslash scale$$
$$x = [x_1, x_2]$$

Each affine coupling layer transforms only a subset of channels at a time, while the remaining channels pass unchanged. However, because the output is always concatenated as $[y_1, y_2]$ and the next layer again splits it by chunk(2), the identity and transformed roles rotate across layers. Consequently, over multiple layers, all channels are eventually transformed.

**Flow Composition** We stack multiple coupling layers to construct an invertible transformation where the overall flow is denoted $z = f_\phi(x)$ and its inverse $x = f_\phi^{-1}(z)$ reconstructs the input. For images we use bounded scaling to stabilize color shifts

$$z^{(0)} = x$$

$$z^{(i+1)} = CouplingLayer_i(z^{(i)}) \text{ for } i = 0, 1, ..., L - 1$$

$$z = k \odot z^{(L)} \text{ where learnable global scaling factor } k = 1 + tanh(\theta) \in \mathbb{R}^{1 \times C \times 1 \times 1}$$

**Stochastic Perturbation Function** $S_\theta$ A learnable noise generator $S_\theta$ is applied to the latent representation to produce semantic-preserving stochastic perturbations

$$\varepsilon = S_\theta(z) + \delta, \qquad \delta \sim N(0, I)$$

The perturbed latent representation becomes $z + \alpha \cdot \varepsilon$ which is decoded back to the image space via the inverse flow.

$$x' = f_\phi^{-1}(f_\phi(x) + \alpha \cdot \varepsilon) = f_\phi^{-1}(z + \alpha \cdot \varepsilon), \qquad \alpha = 1$$

## C.2 Flow-based model for Speech Augmentation

**1D Coupling Layer** Given input $x \in \mathbb{R}^{B \times 1 \times T}$, we split it along the temporal dimension into even and odd time steps.

$$x_{\text{even}} = x[:, :, 0 :: 2] \in \mathbb{R}^{B \times 1 \times \frac{T}{2}}, \qquad x_{\text{odd}} = x[:, :, 1 :: 2]$$

$$\text{where } x_{\text{even}} \in \mathbb{R}^{B \times 1 \times T}, \qquad x_{\text{odd}} \in \mathbb{R}^{B \times 1 \times T}$$

$x_{\text{odd}}$ be a transformed part conditioned by the conditioning part $x_{\text{even}}$.

The affine transformation parameters be computed as

$$[shift, log\, scale] = chunk(f(x_{\text{even}}), 2, \ \dim = 1)$$

$$shift \in \mathbb{R}^{B \times 1 \times \frac{T}{2}}, \qquad log\, scale \in \mathbb{R}^{B \times 1 \times \frac{T}{2}}$$

Then,

$$scale \leftarrow exp(clamp(log\, scale, -2, 2))$$

The transformed output becomes

$$y_{\text{even}} = x_{\text{even}}$$
$$y_{\text{odd}} = x_{\text{odd}} \odot scale + shift$$
$$y = [y_{\text{even}}, y_{\text{odd}}]$$

This ensures invertibility

$$x_{\text{even}} = y_{\text{even}}$$
$$x_{\text{odd}} = (y_{\text{odd}} - shift) \oslash scale$$
$$x = [x_{\text{even}}, x_{\text{odd}}]$$

We insert and invertible $1 \times 1$ convolution for scale gain.

$$z = W * x, \qquad W \in \mathbb{R}^{1 \times 1}, \quad W \text{ initialized orthogonal}$$

The inverse transformation is

$$x = W^{-1} * z$$

**Flow Composition** We stack multiple coupling layers and invertible $1 \times 1$ convolutions to construct an overall invertible transformation. The forward mapping is denoted $z = f_\phi(x)$ and the inverse $x = f_\phi^{-1}(z)$ reconstructs the input.

$$z^{(0)} = x$$

$$z^{(i+1)} = \text{Layer}_i(z^{(i)}), \quad i = 0, 1, \ldots, L - 1$$

where each $\text{Layer}_i$ alternates an invertible $1 \times 1$ convolution and a temporal affine coupling block.

$$z = k \odot z^{(L)} \text{ where learnable global scaling factor } k = exp(\theta) \in \mathbb{R}^{1 \times 1 \times 1}$$

**Stochastic Perturbation Function** $S_\theta$ A learnable noise generator $S_\theta$ is applied to the latent representation to produce semantic-preserving stochastic perturbations

$$\varepsilon = S_\theta(z) + \delta, \qquad \delta \sim N(0, I)$$

The perturbed latent representation becomes $z + \alpha \cdot \varepsilon$ which is decoded back to the image space via the inverse flow.

$$x' = f_\phi^{-1}(f_\phi(x) + \alpha \cdot \varepsilon) = f_\phi^{-1}(z + \alpha \cdot \varepsilon), \qquad \alpha = 0.05$$

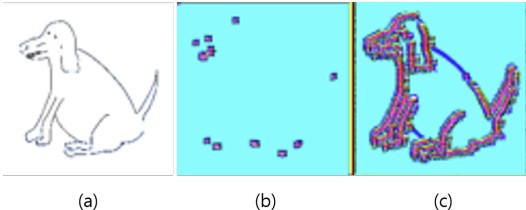

(a)                 (b)                 (c)

Figure 5: (a) original image, (b) Semantically distorted augmentation, (c) Semantic preserving augmentation

## D    SEMANTIC DRIFT ANALYSIS ACROSS PERTURBATION LEVELS

This section provides additional qualitative and quantitative analysis of semantic drift that may occur when the perturbation strength $K$ becomes excessively large. Semantic drift refers to cases where the augmented sample no longer preserves the class-defining structure of the original image. Figure 5 shows representative examples of semantic drift at high perturbation levels. To quantify how often such drift occurs, we manually evaluated samples generated at different perturbation levels. No semantic drift was observed for small perturbations ($K \leq 3$). As $K$ increases, drift begins to appear gradually. 2 out of 5 samples at $K = 5$, 3 out of 10 at $K = 10$, 4 out of 15 at $K = 15$, and 6 out of 20 at $K = 20$. This finding confirm that SLNP maintains semantic consistency for moderate perturbation strengths, and semantic drift emerges when perturbations exceed the stable vicinal region.

