# OpenReview forum: "Multi-Level Stochastic Latent Noise Perturbation for Single Domain Generalization"
_ICLR.cc/2026/Conference — ICLR 2026 Conference Withdrawn Submission_

### Official Review · Reviewer_fJ5b · 2025-10-20

**Soundness:** 3
**Presentation:** 2
**Contribution:** 3
**Rating:** 6
**Confidence:** 4

**Summary:**

This paper addresses a fundamental yet challenging problem, namely single domain generalization (single-DG). Existing approaches based on data augmentation typically face a trade-off between semantic consistency and data diversity. To overcome this limitation, the authors propose a new strategy termed Stochastic Latent Noise Perturbation (SLNP), which enhances generalization by introducing stochastic perturbations in the latent space. SLNP is designed to be a pluggable module applicable to existing single-DG frameworks. Experiments conducted on both image and audio datasets demonstrate the effectiveness of the proposed method.

**Strengths:**

-	The paper is well organized and clearly written.

-	The proposed strategy is novel and interesting. It introduces distribution-based augmentations in a modality-agnostic manner, which is conceptually elegant.

-	Experiments on both vision and speech datasets provide evidence for the generality of the proposed approach.

**Weaknesses:**

-	The experimental scope is relatively limited. The evaluation focuses mainly on two image datasets (PACS and CIFAR-10-C) in the vision domain and one audio dataset (TAU) in the audio domain. Since a key argument of the paper is the modality-agnostic property of SLNP, it would be more convincing to include experiments on additional modalities. For instance, MAD [1] evaluates performance on 1D, 2D, and 3D classification as well as 2D dense segmentation tasks, which provides a stronger demonstration of generality.

-	The analysis section is rather brief, containing only an ablation study. It would strengthen the paper to include more comprehensive analyses, such as hyperparameter sensitivity, visualization, or component-level generalization analysis [2].

-	The result discussion could provide deeper insights, particularly for cases where the vanilla performance is inferior to the baseline (e.g., Table 3). It would be helpful to explain potential reasons and implications for such cases.


**References:**

[1] Modality-agnostic debiasing for single domain generalization. CVPR, 2023.

[2] High-frequency component helps explain the generalization of convolutional neural networks. CVPR, 2020.

**Questions:**

Please refer to the weakness section.

---

> ### Author Response · Authors · 2025-11-21
>
> Dear Reviewer fJ5b,
>
> Thank you for your review and for pointing out the issues. We want to report additional analysis and ablation studies addressing your key concerns.
>
> 1. The experimental scope is relatively limited. The evaluation focuses mainly on two image datasets (PACS and CIFAR-10-C) in the vision domain and one audio dataset (TAU) in the audio domain. Since a key argument of the paper is the modality-agnostic property of SLNP, it would be more convincing to include experiments on additional modalities. For instance, MAD [1] evaluates performance on 1D, 2D, and 3D classification as well as 2D dense segmentation tasks, which provides a stronger demonstration of generality.
>
> Since SLNP is a modality-agnostic augmentation module, its validation requires showing that the same latent perturbation process remains effective across different input modalities. Unlike classifier redesign methods such as MAD, augmentation modules are more sensitive to input structure, so extending them across modalities is a more challenging and meaningful test of generality. For this reason, we focused on representative vision and audio tasks to demonstrate cross-modal applicability. Although extending SNLP to more modalities and tasks is feasible and will be discussed as future work, our experimental scope reflects this methodological focus on input-driven modality-agnostic augmentation, which is yet unexplored, rather than benchmarking a new classifier architecture across all task types.
>
> 2. The analysis section is rather brief, containing only an ablation study. It would strengthen the paper to include more comprehensive analyses, such as hyperparameter sensitivity, visualization, or component-level generalization analysis [2].
>
> Regarding the depth of analysis, we have updated the paper to include sensitivity analyses of $\lambda_1$, $\lambda_2$, $\alpha$, and perturbation level $K$, semantic drift analysis, and metric replacement results. These analyses empirically support that SLNP preserves semantics while expanding domain-specific variability.
>
> 3. The result discussion could provide deeper insights, particularly for cases where the vanilla performance is inferior to the baseline (e.g., Table 3). It would be helpful to explain potential reasons and implications for such cases.
>
> Although SLNP alone performs slightly below the vanilla baseline in Table 3, this is expected because SLNP is designed as part of an augmentation-normalization framework, not as a standalone augmentation method. As shown in related works, augmentation increases diversity while normalization stabilizes semantics, and the two are complementary by design. Accordingly, the combination of SLNP and normalization consistently outperforms both vanilla and normalization alone across datasets.

---

> > ### Comment · Reviewer_fJ5b · 2025-11-24
> >
> > I would like to appreciate the authors’ response to my concerns. However, it does not adequately address them.
> >
> > First, since SLNP is modality-agnostic, it should be evaluated across diverse modalities—such as in MAD, which includes 1D, 2D, and 3D inputs. Without such experiments, the overall effectiveness of SLNP remains questionable.
> >
> > Second, the analysis is insufficient. Beyond ablation studies, more comprehensive quantitative and qualitative analyses are needed to demonstrate that SLNP helps baseline models learn more semantic-rich representations and improves generalization, as shown in prior work like MAD.
> >
> > In summary, a major revision and extension of the manuscript are recommended.

---

### Official Review · Reviewer_W9RJ · 2025-10-26

**Soundness:** 3
**Presentation:** 2
**Contribution:** 2
**Rating:** 2
**Confidence:** 4

**Summary:**

This paper introduces a Stochastic Latent Noise Perturbation module for single-domain generalization.

**Strengths:**

The proposed method perturbs latent representations using a flow-based model under multiple Maximum Mean Discrepancy constraints that are automatically computed from intra-class and inter-class statistics of the source domain. By doing so, it aims to generate diverse yet semantically consistent augmented samples without adversarial training or additional loss terms. The approach is designed to be classifier-independent and applicable across modalities, demonstrated on both vision (PACS, CIFAR-10-C) and speech (TAU Urban Acoustic Scenes) datasets. Experimental results show that SLNP improves robustness and complements normalization-based methods for better generalization to unseen domains.

**Weaknesses:**

1. **Lack of theoretical justification for stochastic latent perturbation's effectiveness in generalization**

It is well known that introducing perturbations can enhance a model’s generalizability. The paper does not clearly explain (1) the unique advantages of the proposed stochastic latent perturbation and (2) the underlying reasons why it can effectively enhance generalization. The effectiveness of the noise injection remains empirically observed rather than theoretically supported.

2. **The calculation method for $\epsilon$ risks circular logic.**

Both $\xi_{inter}$ and $\xi_{intra}$ rely on statistics of class distributions from the training data, which themselves can be affected by data scale and noise. If the source domain's inter-class distance is minimal or the intra-class variance is large, the calculation of $\epsilon_{max}$ may become unstable or fail.

3. **The design of modal adaptation mechanisms is not uniform.**

The paper claims that "the same framework can handle both vision and speech", but the speech part modifies the structure (1D flow, temporal splitting) and adjusts the hyperparameters (K=3 instead of 15), indicating that the method still requires manual parameter adjustment in different modalities and is not a truly unified algorithm.

4. **Limited experimental scale and improvement.**

(1) The improvement on PACS is marginal, only 0.06% (70.22% vs 70.16%), which is almost within the margin of error.

(2) On the speech task, using the method alone actually degrades performance (35.25% → 31.82%).

(3) This suggests the method's intrinsic contribution is limited and lacks statistical significance verification (e.g., t-tests, confidence intervals).

5. **Insufficient ablation studies**

The study only presents a sensitivity analysis for the parameter K, lacking ablations for the following critical factors:

(1) Whether multi-level MMD is truly necessary (single-level vs. multi-level).

(2) The impact of $\lambda_1$, $\lambda_2$, and $\alpha$ on performance.

(3) Whether the method remains effective if the MMD metric is replaced (e.g., with Wasserstein distance or cosine similarity).

6. **Imbalance between vision and speech Tasks.**

(1) While the vision component is relatively well-experimented (two datasets), the speech component relies on only one small dataset (TAU-ASC).

(2) The lack of cross-task validation (e.g., speech recognition, emotion classification) weakens the argument for the method's cross-modal generalization capability.

**Questions:**

Refer to the Weaknesses above.

---

> ### Author Response · Authors · 2025-11-21
>
> Dear Reviewer W9RJ,
>
> Thank you for your review and for pointing out the issues. We want to report additional theoretical analysis and ablation studies addressing your key concerns.
>
> 1. Lack of theoretical justification for stochastic latent perturbations in generalization
>
> - Our modular augmentation method operates without end-to-end adversarial training, expanding diversity while preserving semantics through an MMD-based constraint. Additionally, our framework directly transfers to different data modalities (e.g., images and speech), enabling a modality-agnostic perspective on single-domain generalization.
> - We perturbed the data in the latent space rather than the raw data space. Latent-space augmentations such as Manifold Mixup (Verma et al., 2019) theoretically encourage flatter decision boundaries, which correlate with improved generalization.
>
> 2. The calculation method for risks circular logic.
>
> Such a case would imply that the domain is already diverse, leading to a smaller epsilon_max. This will result in the MMD gap becoming small in Eqn 3, which means x and x’ are ‘similar’ to each other. The rationale is that since the domain is diverse, it’s okay to let the augmented samples (x’) be similar to the originals (x).
> On the other hand, small inter-class distance and large intra-class distance also mean that the classification itself is difficult. This would constitute an ill-posed problem to begin with.
>
> 3. The design of modal adaptation mechanisms is not uniform.
>
> It is the overall framework that we claim to be modality-agnostic, not the specific architecture. Different modalities need different processing steps. If one were to claim a modality-agnostic architecture that doesn’t require structural fine-tuning, then at least the inputs would need tailored feature extraction. This part is unavoidable.
>
> 4. Limited experimental scale and improvement.
>
> PACS improvement is small but consistent, and on TAU-UAS the performance drop occurs only when SLNP is used alone.  However, we insist that our augmentation method is not intended to function as a standalone classifier, but as a classifier-independent augmentation module within an augmentation + normalization framework. When used as designed, in combination with normalization, SLNP reliably improves performance on both vision and speech tasks. Therefore, interpreting the isolated SLNP result as evidence of limited contribution overlooks the method’s intended role and its clear complementary effect, which consistently strengthens normalization-based generalization across modalities.
>
> 5. Insufficient ablation studies
>
> We conduct a parameter sensitivity analysis on CIFAR-10-C by varying both 1(noise magnitude maximization) and  2(mmd constraint). Specifically, we vary both parameters {0.05, 0.1, 1.0} while keeping all other components fixed. The resulting accuracy yields a small variance of 82.78  0.50, indicating that the proposed SLNP module is highly robust to the choice of loss balancing coefficients. Additionally, the scalar multiplied by the random noise is fixed throughout all experiments. It serves only as a stability factor to prevent excessive noise injection during the early stages of flow inversion.
>
> To verify whether our method depends critically on the choice of MMD, we replaced the MMD constraint with two alternative discrepancy measures commonly used in distributional alignment: Wasserstein distance and cosine similarity. The resulting accuracies of the sketch domain in PACS are MMD: 62.49, Wasserstein distance: 59.67, and Cosine similarity: 59.75. Although our method is effective with other discrepancy measures, MMD offers the best combination, which is why we use it as the default metric in our framework.
>
> 6. Imbalance between vision and speech Tasks.
>
> - In vision only SDG, it is standard to evaluate on 3 vision benchmarks. However, we aim to demonstrate that SLNP is modality-agnostic, functioning across both vision and speech. Because we allocate experimental coverage across two modalities rather than three vision datasets, the number of datasets per modality is smaller, but the overall evaluation scope becomes broader.
> - We agree that evaluating additional tasks would further broaden the empirical scope. We consider this a promising direction for future work, but it is beyond the intended focus of demonstrating modality-agnostic perturbation within the classification setting.
>
>
> Verma, V., Lamb, A., Beckham, C., Najafi, A., Mitliagkas, I., Lopez-Paz, D., & Bengio, Y. (2019, May). Manifold mixup: Better representations by interpolating hidden states. In International conference on machine learning (pp. 6438-6447). PMLR.

---

> > ### Comment · Reviewer_W9RJ · 2025-11-24
> >
> > I would like to appreciate the authors’ response to my concerns. However, my core concern (the first comment) still remains insufficiently addressed.
> >
> > Broadly speaking, the idea that adding noise can improve generalization is not new; this phenomenon has been reported in many prior works, as the authors mentioned above. Frankly, if the main contribution of this paper is merely injecting random noise into the features, it does not present a particularly distinctive novelty. A theoretical explanation for why this strategy works would meaningfully distinguish the paper from earlier empirical studies. As it stands, it is still unclear why adding random noise should reduce domain discrepancy.

---

> > > ### Author Response · Authors · 2025-12-02
> > >
> > > Thank you for the comment. We agree that simply adding random noise is not novel; however, our method is fundamentally different from unconstrained noise injection. The perturbations in SLNP are learned through a normalizing flow trained on the source-domain feature distribution and are further restricted by a divergence constraint (MMD), ensuring that all augmented samples remain within a semantically coherent vicinal neighborhood of the original features, which aligns directly with the vicinal risk minimization principle. Because the flow captures the geometry of the source distribution, the perturbations expand variability primarily along domain-variant directions while preserving class semantics, which theoretically and empirically reduces domain discrepancy.

---

### Official Review · Reviewer_44tk · 2025-10-29

**Soundness:** 2
**Presentation:** 1
**Contribution:** 2
**Rating:** 2
**Confidence:** 3

**Summary:**

This paper addresses Single-Domain Generalization (SDG) problem, where a model trained on a single domain is to generalize to unseen domains. Specifically, it proposes a new augmentation framework that trains a flow-based model under a multi-level Maximum Mean Discrepancy constraint and perturbs latent representations in a semantically consistent way. Experiments on both image and speech benchmarks demonstrate its superior generalization performance.

**Strengths:**

1. The idea of using a flow-based model and injecting noise in the latent space is interesting, as it expands diversity without risking semantic degradation.

2. The proposed module is independent of the main network and can serve as a plug-and-play data augmentation component, making it easily combinable with other methods.

**Weaknesses:**

1. The paper is not easy to follow. First, the organization is unclear: there is no summary of contributions, and the first two paragraphs of Section 3.4 read more like related work, yet they are placed under the Proposed Method section. Moreover, the paper lacks a framework figure and a brief introduction to RealNVP[1], which makes it difficult for readers unfamiliar with RealNVP to understand the method. Second, the logical consistency in some sentences is weak. For instance, in lines 265–267, the sentence first mentions “both methods” but then provides a reason that applies only to ADA, creating a logical disconnect between cause and subject. Additionally, in line 268, ASR-Norm is mentioned for the first time without any reference. Third, in line 205, Third, in line 205, the symbols $k$ and $x'$ in $E_x [k(x,x)]-E_(x,x') [k(x,x')]$ are coincided with those in Eq. (3), but the two expressions have different meanings, which may confuse readers.

2. In line 153, the paper states that “a learnable per-channel global scale is applied at the end of the flow.” However, the motivation for this design choice is not clearly explained. Could the authors provide more intuitive reasoning for this?

3. In line 271, the paper claims that the method “perturbs only domain-specific factors.” However, there is no disentanglement constraint or explicit mechanism to separate domain-specific and semantic factors in the latent space. Could the authors clarify how the model ensures that only domain-specific factors are perturbed?

4. For the image datasets, the authors evaluate their method on only two small datasets and do not include comparisons with recent works. Since the paper was submitted in September, it should consider more up-to-date baselines, such as SAC [2]. Could the authors conduct additional experiments on a larger benchmark, such as DomainNet [3], and compare their method with SAC to provide a more comprehensive evaluation?

5. For ablation study, the paper introduces two loss terms in Eq. (3); however, there is no ablation analysis to evaluate their individual contributions. Could the authors include additional experiments to analyze the effect of each loss component on model performance?

[1] Dinh, Laurent, et al. "Density estimation using Real NVP." International Conference on Learning Representations. 2017.

[2] Zhang, Zhen, et al. "Split-and-Combine: Enhancing Style Augmentation for Single Domain Generalization." Proceedings of the IEEE/CVF International Conference on Computer Vision. 2025.

[3] Peng, Xingchao, et al. "Moment matching for multi-source domain adaptation." Proceedings of the IEEE/CVF international conference on computer vision. 2019.

**Questions:**

Please see the above weaknesses. If you can conduct additional experiments to further evaluate your method, I would be willing to raise my score.

---

> ### Author Response · Authors · 2025-11-21
>
> Dear Reviewer 44tk,
>
> Thank you for your review and for pointing out the issues. We want to clarify the issues and additional ablation studies addressing your key concerns.
>
> 1. Organization and clarity issue
>
> We have revised the paper to address your clarity issues. We added a summary of contributions, reorganized Section 3.4 to related work, and included both a framework figure and a brief introduction to RealNVP for readability. We also corrected the logical inconsistency, properly introduced ASR-Norm with citation, and fixed the notation conflict to avoid confusion.
>
> 2. In line 153, the paper states that “a learnable per-channel global scale is applied at the end of the flow.” However, the motivation for this design choice is not clearly explained. Could the authors provide more intuitive reasoning for this?
>
> The per-channel global scale adjusts the dynamic range of latent channels after the coupling layers. Since the coupling log-scales are clamped for numerical stability in the experiment, they cannot freely control global variance. Thus, the global scale ensures that the latent channel has an appropriate magnitude before noise injection, preventing any channel from dominating the perturbation and helping the flow decoder reconstruct semantically valid on-manifold samples.
>
> 3. In line 271, the paper claims that the method “perturbs only domain-specific factors.” However, there is no disentanglement constraint or explicit mechanism to separate domain-specific and semantic factors in the latent space. Could the authors clarify how the model ensures that only domain-specific factors are perturbed?
>
> I agree that the phrasing “perturbs only domain-specific factors” is overly strong and may imply explicit disentanglement. (Revise to “perturbs primarily domain-specific factors) Our method doesn’t implement an explicit semantic-style separation; instead, we insist implicit effect due to the interaction between the MMD constraint and the flow-based latent representation. Perturbations that distort semantics are naturally suppressed by the MMD constraint.
>
> 4. For the image datasets, the authors evaluate their method on only two small datasets and do not include comparisons with recent works. Since the paper was submitted in September, it should consider more up-to-date baselines, such as SAC [2]. Could the authors conduct additional experiments on a larger benchmark, such as DomainNet [3], and compare their method with SAC to provide a more comprehensive evaluation?
>
> Although SAC is a recent baseline, since the code is not publicly available and the experiment setting is different from ours, we compared our method against LEAwareSGD, another up-to-date SGD baseline.
>
> 5. For the ablation study, the paper introduces two loss terms in Eq. (3); however, there is no ablation analysis to evaluate their individual contributions. Could the authors include additional experiments to analyze the effect of each loss component on model performance?
>
> We conduct a parameter sensitivity analysis on CIFAR-10-C by varying both $\lambda_1$(noise magnitude maximization) and  $\lambda_2$(mmd constraint). Specifically, we vary both parameters $\in$ {0.05, 0.1, 1.0} while keeping all other components fixed. The resulting accuracy yields a small variance of 82.78 $\pm$ 0.50, indicating that the proposed SLNP module is highly robust to the choice of loss balancing coefficients. Additionally $\alpha$, the scalar multiplied by the random noise is fixed throughout all experiments. It serves only as a stability factor to prevent excessive noise injection during the early stages of flow inversion.

---

### Official Review · Reviewer_wyCr · 2025-11-01

**Soundness:** 2
**Presentation:** 2
**Contribution:** 2
**Rating:** 2
**Confidence:** 3

**Summary:**

The paper proposes a modality-agnostic augmentation module for single-domain generalization. Inputs are mapped through an invertible RealNVP-style flow, perturbed with stochastic latent noise, and reconstructed; perturbation magnitude is governed by K automatically derived MMD thresholds computed from intra/inter-class statistics, with a curriculum-like schedule. The module is pretrained outside the main training loop and then used as a plug-in for both images and speech; sensitivity to K is analyzed.

**Strengths:**

1. The method presents a general mechanism suitable for both speech and image data. Implementation details are clearly outlined for 1D and 2D flows.

2. Automatic thresholding is achieved using intra- and inter-class MMD. The analysis includes a sensitivity study on the parameter K.

**Weaknesses:**

1. The empirical evidence needs strengthening. A direct, quantitative comparison against strong SDG baselines across multiple datasets is missing. This comparison should report mean±std and effect sizes, as several claims currently rely on qualitative statements.

2. The RealNVP-style flows introduce additional training and inference costs. The practical impact on wall-clock time is not discussed, nor is it clear if cheaper perturbations could yield similar performance gains.

**Questions:**

1. Provide a summary table with mean±std results across multiple runs for PACS, Digits, CIFAR-10-C, and TAU-UAS. The table should compare (a) backbone only, (b) best normalization baseline, (c) SLNP, and (d) SLNP+norm.

2. Quantify the training and inference overhead from the flow module, both during pretraining and in the main loop. Report this in GPU hours and images/second.

3. Show examples of failure cases where a large K causes semantic drift. Also, indicate how frequently this occurs in practice.

---

> ### Author Response · Authors · 2025-11-21
>
> Dear Reviewer wyCr,
>
> Thank you for your review and for acknowledging our work's experimental validation. We want to report additional experiments for your key concerns.
>
> 1. Provide a summary table with mean±std results across multiple runs for PACS, Digits, CIFAR-10-C, and TAU-UAS. The table should compare (a) backbone only, (b) best normalization baseline, (c) SLNP, and (d) SLNP+norm.
>
> We provide a complete summary table reporting mean ± std across three random seeds for PACS, CIFAR-10-C, and TAU-UAS. This ensures that all benchmarks follow a consistent multi-seed evaluation protocol for fair and reproducible comparison.
>
> 2. Quantify the training and inference overhead from the flow module, both during pretraining and in the main loop. Report this in GPU hours and images/second.
>
> We provide detailed measurements for both the pretraining stage and the main training loop. All experiments were conducted on a single NVIDIA GeForce RTX 3090 GPU, and the computational overhead was quantified onthe  CIFAR-10-C dataset. The augmentation precomputation consists of pool feature extraction, multi-level epsilon computation, and flow-based perturbation generation. Their measured wall-clock times are 54.97 sec, 7.94 sec, 441.11 sec, respectively. The total precomputation overhead is 0.14 GPU hours. This preprocessing is performed once per dataset and does not occur during the main training loop.
>
> After epsilon-wise flow training is completed, the flow model is not trained further. It is only used in forward mode to generate perturbations. The resulting training throughput is 8,200-10,600 images/sec.
>
> The full CIFAR-10-C experiment, including all components, required 58.77 GPU hours, including one-time precomputation cost.
>
> 3. Show examples of failure cases where a large K causes semantic drift. Also, indicate how frequently this occurs in practice.
>
> To clarify the effect of large $K$, we conducted a semantic distortion analysis by manually inspecting augmented samples and identifying cases where class-defining structures were noticeably corrupted (e.g. missing lines, shape collapse). We provide an example in the supplementary material (appendix) and analyze the semantic distortion. No drift occurred for very small perturbations ($K\leq3$), while the drift frequency gradually increased for larger $K$. This observation supports our interpretation that increasing $K$ enhances diversity up to a point, but overly large $K$ might introduce a higher likelihood of semantic drift. This aligns with our experimental outcome that $K=15$ provides the best balance between diversity and semantic preservation.

---

### Comment · Area_Chair_wzuy · 2025-11-27
**Reminder: Engage in Discussions and Finalize Your Review**

Dear Reviewers,

Thank you for your valuable reviews. With the Reviewer-Author Discussions deadline approaching, please take a moment to read the authors’ rebuttal and the other reviewers’ feedback, and participate in the discussions and respond to the authors. Finally, be sure to complete the “Final Justification” text box and update your “Rating” as needed. Your contribution is greatly appreciated. I will flag irresponsible (final) reviews and/or any reviewers not participating in discussions.

Reviewers are expected to stay engaged in discussions, initiate them, respond to authors’ rebuttal, ask questions, and listen to answers to help clarify remaining issues.

It is not OK to stay quiet.

It is not OK to leave discussions till the last moment.

If authors have resolved your (rebuttal) questions, do tell them so.

If authors have not resolved your (rebuttal) questions, do tell them so too.

Thanks,

AC

---

### Note · Authors · 2026-01-11

I have read and agree with the venue's withdrawal policy on behalf of myself and my co-authors.